# Improved Biotransformation of Platycoside E into Deapiose-Xylosylated Platycodin D by Cytolase PCL5 under High Hydrostatic Pressure

**Kyung-Chul Shin [1], Min-Ju Seo [2], Yu Jin Oh [3], Dae Wook Kim [3], Chae Sun Na [3,*] and Yeong-Su Kim [3,*]**

1   Department of Integrative Bioscience and Biotechnology, Konkuk University, Seoul 05029, Korea; hidex2@konkuk.ac.kr
2   Department of Biochemistry, Molecular Biology & Biophysics, University of Minnesota, 140 Gortner Laboratory, 1479 Gortner Avenue, Saint Paul, MN 55108, USA; seo00056@umn.edu
3   Wild Plants and Seeds Conservation Department, Baekdudaegan National Arboretum, Bonghwa 36209, Korea; oyj0705@koagi.or.kr (Y.J.O.); dwking@koagi.or.kr (D.W.K.)
*   Correspondence: chaesun.na@koagi.or.kr (C.S.N.); yskim@koagi.or.kr (Y.-S.K.)

**Abstract:** Platycosides are the functional saponins present in balloon flowers that exert diverse biological effects, and which can be further improved by their deglycosylation. Deapiose-xylosylated platycodin D, which is absent in balloon flowers, can be generated only by cytolase PCL5 by acting on platycoside E. To improve cytolase PCL5-catalyzed production of deapiose-xylosylated platycodin D from platycoside E, we explored the use of high hydrostatic pressure (HHP). At an HHP of 150 MPa, the optimal temperature of cytolase PCL5 activity for converting platycoside E into deapiose-xylosylated platycodin D shifted from 50 to 55 °C, and increased the activity and stability of the enzyme by 5- and 4.9-fold, respectively. Under HHP, the enzyme completely converted 1 mM platycoside E into deapiose-xylosylated platycodin D within 4 h, with a 3.75-fold higher productivity than that under atmospheric pressure. Our results suggest that the application of HHP is a potential method for the economical production of platycosides and enzyme-catalyzed biotransformation of functional saponins.

**Keywords:** platycoside; cytolase PCL5; balloon flower; deglycosylation; high hydrostatic pressure

## 1. Introduction

Balloon flowers have been traditionally used as a medicinal diet in Northeast Asia, due to their efficacy against cold, cough, tonsillitis, sore throat, bronchitis, and chest congestion [1]. They contain saponins called platycosides, which have been attracting increasing interest among researchers because of their nutraceutical and pharmacological activities, including anti-inflammatory [2,3], anti-oxidant [4,5], antitumor [6,7], anti-obesity [8,9], and immunoregulatory effects [10,11].

Platycosides are composed of a pentacyclic triterpene aglycone with two sugar chains consisting of glucose residues at C-3 and oligosaccharides of arabinose, rhamnose, xylose, and apiose at the C-28 position (Figure 1). The biological activity of saponins is increased by deglycosylation, which enhances their absorption in the human gut due to lower molecular size and better hydrophobicity [12,13]. Thus, studies have explored saponin deglycosylation using heat [14], acid [15], and enzyme treatments [16]. Since enzymatic bioconversion of saponins has high selectivity, studies on the platycoside conversion have been carried out using various enzymes such as β-glucosidases from *Aspergillus usamii* [17], *Caldicellulosiruptor bescii* [18], *Caldicellulosiruptor owensensis* [19], *Dictyoglomus turgidum* [20], and cellulase [21], pectinase [22], and cytolase PCL5 [23]. Among these, cytolase PCL5 is the only enzyme that can hydrolyze the outer glucose at C-3 and xylose and apiose at C-28 in platycoside E, the most abundant saponin in balloon flower (Figure 1).

| Platycoside | $R_1$ | $R_2$ |
|---|---|---|
| PE | Glc(6→1)Glc(6→1)Glc | Arap(3→1)Rha(4→1)Xyl(3→1)Api |
| PD$_3$ | Glc(6→1)Glc | Arap(3→1)Rha(4→1)Xyl(3→1)Api |
| PD | Glc | Arap(3→1)Rha(4→1)Xyl(3→1)Api |
| Deapi-PD | Glc | Arap(3→1)Rha(4→1)Xyl |
| Deapi-xyl-PD | Glc | Arap(3→1)Rha |

**Hydrolytic pathway by Cytolase PCL5**

**Figure 1.** Chemical structure and hydrolytic pathway of platycoside E into deapiose-xylosylated platycodin D via intermediates platycodin D$_3$, platycodin D, and deapiosylated platycodin D catalyzed by cytolase PCL5. PE, platycoside E; PD$_3$, platycodin D$_3$; PD, platycodin D; Deapi-, deapisoylated; Deapi-xyl-, deapiose-xylosylated; Glu, β-D-glucopyranosyl-; Arap, α-L-arabinopyranosyl-; Rha, α-L-rhamnopyranosyl-; Xyl, β-D-xylopyranosyl-; and Api, β-D-apiofuranosyl-.

High hydrostatic pressure (HHP) processing has been used to preserve and sterilize food products by maintaining very high-pressure conditions, which leads to the inactivation of some microorganisms and enzymes. On the contrary, within a certain range, HHP can also improve the stability and activity of several enzymes such as viscozyme, pectinase, cellulase, amylase, α-L-arabinofuranosidase, and α-L-rhamnosidase [24]. However, HHP has never been studied for improving the enzymatic conversion of platycosides [25]. In this study, we applied HHP during the bioconversion of platycoside, catalyzed by cytolase PCL5, to enhance the production of deapiose-xylosylated platycodin D from platycoside E.

## 2. Materials and Methods

### 2.1. Materials

Cytolase PCL5 was purchased from DSM Food Specialties (Heerlen, The Netherlands). Platycoside E, platycodin D3, platycodin D, and deapiosylated platycodin D were purchased from Ambo Laboratories (Daejeon, Republic of Korea). Deapiose-xylosylated platycodin D was prepared as previously reported [23] and used as a standard. All other reagents were purchased from Sigma-Aldrich (St. Louis, MO, USA).

### 2.2. Enzyme Assay

The activity of cytolase PCL5 was measured in a reaction mixture containing 50 mM citrate/phosphate buffer (pH 5.0), 0.05 mg/mL cytolase PCL5, and 0.4 mM platycoside for 10 min at 50 or 55 °C and at atmospheric pressure (AP, 0.1 MPa) or HHP (150 MPa). The specific activities of cytolase PCL5 for platycosides such as platycoside E, platycodin D3, platycodin D, deapiosylated platycodin D, and deapiose-xylosylated platycodin D were evaluated at various concentrations (0.005–0.5 mg/mL) of the enzyme in order not to hydrolyze more than one sugar. The specific activity was defined as the amount of platycodin D3, platycodin D, deapiosylated platycodin D, or deapiose-xylosylated platycodin D, which was produced from platycoside E, platycodin D3, platycodin D, or deapiosylated platycodin D, respectively, as a product per enzyme amount per unit reaction time.

### 2.3. Optimization of Reaction Conditions

The effect of pressure was evaluated using 0.4 mM platycoside E for 10 min at 50 °C in citrate/phosphate buffer (pH 5.0) under AP (0.1 MPa) and HHP (0.1–400 MPa), using an HHP instrument (TFS-2L, Toyo-Koatsu Innoway Co. Ltd., Hiroshima, Japan). The effects of pH and temperature on the activity of cytolase PCL5 were examined by varying the pH from 4.0 to 7.0 at 50 °C, and by varying the temperature from 40 to 65 °C at a pH of 5.0, respectively, under AP (0.1 MPa) and HHP (150 MPa). The thermostability of cytolase PCL5 was monitored as a function of incubation duration by maintaining the enzyme solutions at 45, 50, 55, 60, and 65 °C in citrate/phosphate buffer (pH 5.0) under AP (0.1 MPa) and HHP (150 MPa). After incubation, the reaction samples were assayed using 0.4 mM platycoside E in citrate/phosphate buffer (pH 5.0) at 55 °C for 10 min.

### 2.4. Bioconversion

The bioconversion of platycoside E to deapiose-xylosylated platycodin D was performed under AP (0.1 MPa) and HHP (150 MPa) for 15 and 5 h, respectively, at 55 °C in 50 mM citrate/phosphate buffer (pH 5.0) containing 0.5 mg/mL Cytolase PCL5 and 1 mM platycoside E. Samples were taken at 5 min, 10 min, 30 min, 1 h, 3 h, 6 h, 9 h, 12 h, and 15 h under AP, and at 5 min, 10 min, 30 min, 1 h, 2 h, 3 h, 4 h, and 5 h under HHP, respectively, and the experiment was performed in triplicate.

### 2.5. HPLC Analysis

To stop the reaction and extract the platycoside, an equal amount of n-butanol was added to the reaction mixture. The n-butanol-soluble fraction of the extract was separated and dried to completely evaporate butanol. The dried residues were dissolved in methanol and analyzed at 203 nm using an HPLC system (Agilent 1100, Santa Clara, CA, USA) equipped with a hydrosphere C18 column (4.6 × 150 mm, 5 µm particle size; YMC, Kyoto, Japan). The column was eluted at a flow rate of 1 mL/min and 30 °C with a gradient of acetonitrile and water from 10:90 to 40:60 for 30 min, from 40:60 to 90:10 for 15 min, from 90:10 to 10:90 for 5 min, and constant at 10:90 for 10 min. The calibration curves relating the logarithmic value of the peak areas to the concentrations of platycosides were drawn using the solutions of platycoside standards (0.2 to 1.0 mM) and the curves were used to determine the concentrations of platycosides.

## 3. Results and Discussion

### 3.1. Effects of Pressure, pH, and Temperature under AP and HHP on Cytolase PCL5 Activity

To determine the appropriate pressure for the reaction, the hydrolytic activity of cytolase PCL5 was evaluated at pressures ranging from 0 to 400 MPa (Figure 2). The relative activity increased to 423% as the pressure increased to 150 MPa, and then decreased to almost 38% at 400 MPa. Therefore, all subsequent experiments were performed under HHP at 150 MPa. Even when HHP was applied to isoquercetin production with $\alpha$-L-rhamnosidase in the previous study, it showed the highest productivity at 150 MPa [25]. However, it showed approximately 2.6-fold higher productivity than that under AP, which was lower than the increase in the hydrolytic activity of cytolase PCL5.

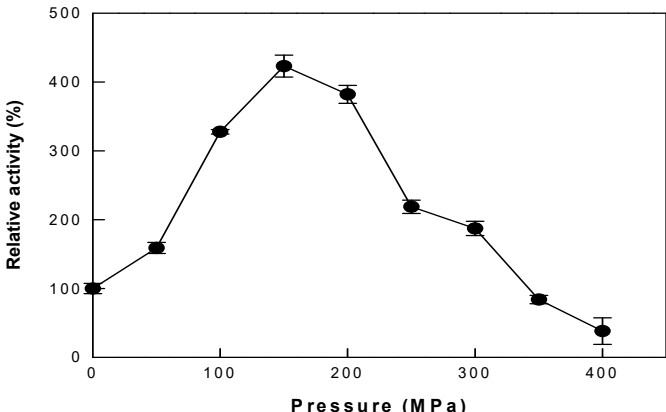

**Figure 2.** The activity of cytolase PCL5 with changes in pressure. Data represent the means of three experiments ± standard deviation.

Optimal pH and temperature for cytolase PCL5 activity on platycoside E have been reported as pH 5.0 and 50 °C [23]. In this study, the maximal activity of cytolase PCL5 was observed at pH 5.0, under both AP and HHP. However, maximum hydrolytic activities were observed at 50 °C and 55 °C under AP and HHP, respectively (Figure 3). At 55 °C, the enzyme activity under HHP was 5-fold higher than that under AP. This result indicates that changes in pressure can affect enzyme activity at a given temperature.

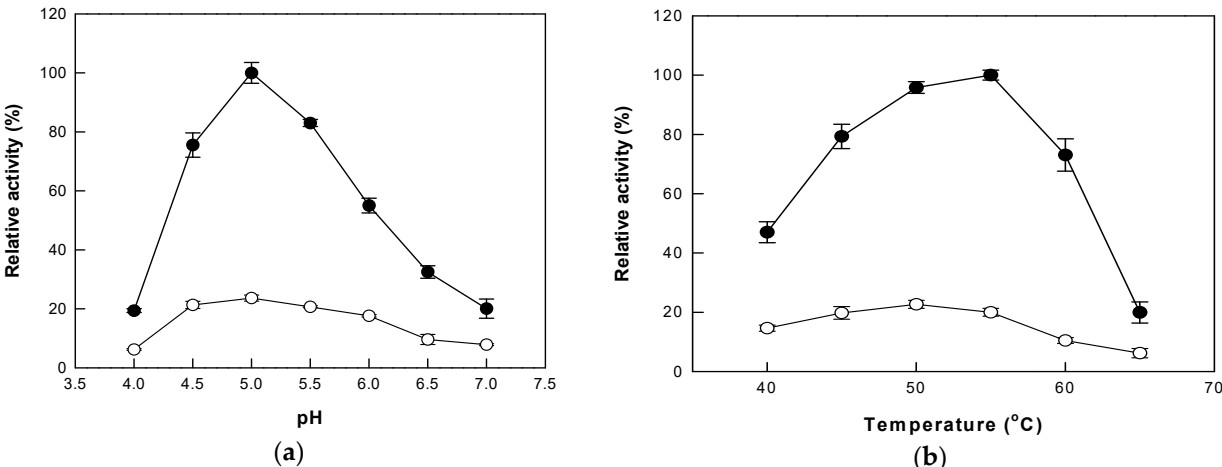

(**a**)
(**b**)

**Figure 3.** The activity of cytolase PCL5 with changes in (**a**) pH and (**b**) temperature under atmospheric pressure (open circle) and high hydrostatic pressure (closed circle). Data represent the means of three experiments ± standard deviation.

### 3.2. Thermal Stability of Cytolase PCL5 under AP and HHP

To accurately evaluate the effects of changing pressure on cytolase PCL5 activity at a given temperature, the thermostability of the enzyme was examined in the range of 45–65 °C under AP and HHP (Figure 4). First-order kinetic reactions were observed for thermal inactivation of cytolase PCL5 and the half-lives of the enzyme were 7.4, 5.1, 2.1, 1.3, and 0.5 h under AP, and 30.1, 21.2, 10.8, 4.8, and 2.2 h under HHP at 45, 50, 55, 60 and 65 °C, respectively. The overall thermal stability of cytolase PCL5 increased under HHP, and the half-life at 55 °C was 4.9-fold higher than that at other temperatures, compared to that under AP, which was higher than the 4-fold increase in thermal stability under HHP in a study regarding the production of isoquercetin [25]. This result was similar to the observation wherein the temperature at which the enzyme showed the highest activity shifted to 55 °C under HHP, which suggested that HHP can simultaneously increase the activity and stability of the enzyme at a specific temperature.

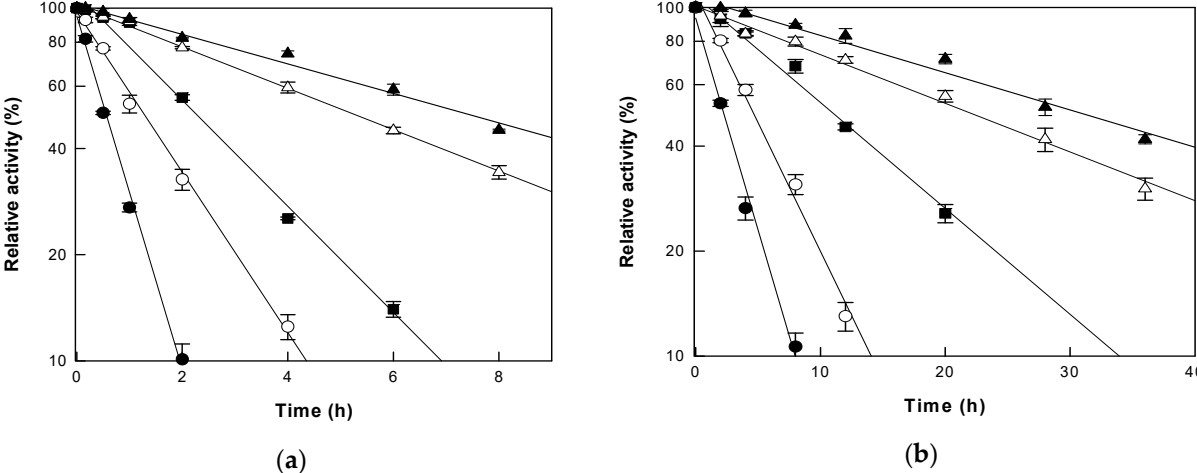

**Figure 4.** Thermal inactivation of cytolase PCL5 under (**a**) atmospheric pressure and (**b**) high hydrostatic pressure. The enzyme was incubated at 45 (closed triangle), 50 (open triangle), 55 (closed square), 60 (open circle), and 65 °C (closed circle). Data represent the means of three experiments ± standard deviation.

### 3.3. Changes in Substrate Specificity with Pressure

The activities of cytolase PCL5 towards platycoside substrates such as platycoside E, platycodin D3, platycodin D, deapiosylated platycodin D, and deapiose-xylosylated platycodin D were compared under AP and HHP (Table 1). The specific activity on the substrates showed the following order, regardless of pressure: platycoside E > platycodin D3 > platycodin D > deapiosylated platycodin D, but the maximum fold increase in enzyme activity under HHP compared to that under AP was observed for deapiosylated platycodin D. These results indicate that high pressure is more effective for difficult to hydrolyze sugars. However, no activity was observed for deapiose-xylosylated platycodin D under either AP or HHP, indicating that even with the maximum hydrolytic activity of cytolase PCL5 under HHP, it was unable to hydrolyze the inner glucose at C-3 and rhamnose at the C-28 positions.

**Table 1.** Specific activities of cytolase PCL5 for different platycoside substrates under AP and HHP.

| Substrate | Specific Activity (nmol/min/mg) | |
| :---: | :---: | :---: |
| | **AP** | **HHP** |
| PE | 15,601.2 ± 50.2 | 48,738.2 ± 101.2 |
| PD$_3$ | 281.2 ± 18.0 | 1056.4 ± 31.5 |
| PD | 35.1 ± 1.8 | 141.5 ± 2.1 |
| Deapi-PD | 15.3 ± 1.3 | 71.9 ± 2.5 |
| Deapi-xyl-PD | ND | ND |

PE, platycoside E; PD$_3$, platycodin D3; PD, platycodin D; Deapi-, deapisoylated; Deapi-xyl-, deapiose-xylosylated; ND, not detected.

### 3.4. Bioconversion of Platycoside E to Deapiose-Xylosylated Platycodin D under AP and HHP

The catalytic bioconversion of platycoside E into deapiose-xylosylated platycodin D was performed with 0.5 mg/mL cytolase PCL5 and 1 mM platycoside E as a substrate, under AP and HHP. The enzyme completely converted platycoside E into deapiose-xylosylated platycodin D within 15 and 4 h with productivities of 66.7 and 250 μM/h under AP and HHP, respectively (Figure 5). The productivity was approximately 3.75-fold higher under HHP than AP, indicating that HHP is much more effective for the production of deapiose-xylosylated platycodin D. Under the HHP condition, platycodin D3 was not detected throughout the reaction due to high hydrolytic activity. During the time-course for the reaction, the highest concentration of deapiosylated platycodin D was lower under HHP (467 μM at 1 h) than AP (609 μM at 6 h). Based on substrate specificity, this result

may be attributable to the higher fold increase in the activity of cytolase PCL5 towards deapiose-xylosylated platycodin D than that toward other platycoside intermediates.

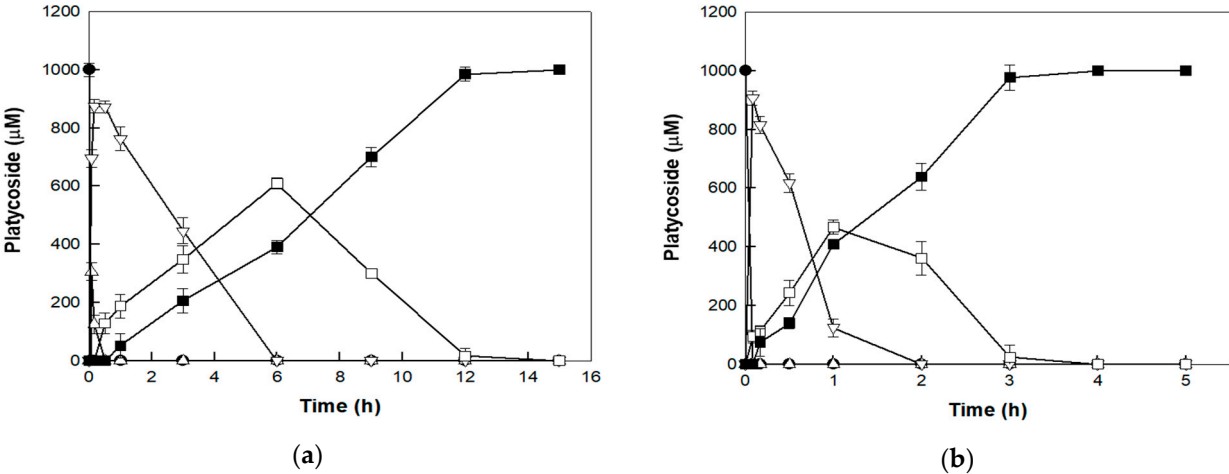

**Figure 5.** Biotransformation of platycoside E (closed circle) into despiose-xylosylated platycodin D (closed square) via platycodin $D_3$ (open triangle), platycodin D (open inverted triangle), and deapiosylated platycodin D (open square) by cytolase PCL5 at (**a**) atmospheric pressure and (**b**) high hydrostatic pressure. Data represent the means of three experiments ± standard deviation.

## 4. Conclusions

Platycoside, the saponin present in the balloon flower, exhibits various pharmacological effects, and its biological activity increases as its sugars are hydrolyzed. Cytolase PCL5 is the only enzyme that can convert platycoside E, the major platycoside in the balloon flower saponin, into deapiose-xylosylated platycodin D. In this study, HHP was applied for the bioconversion of platycoside catalyzed by cytolase PCL5 to improve the production of deapiose-xylosylated platycodin D from platycoside E. The enzyme activity and thermal stability at 55 °C under 150 MPa pressure were approximately 5- and 4.9-fold higher than those under AP, which increased the bioconversion of platycoside E to deapiose-xylosylated platycodin D by 3.75-fold. The application of HHP not only increased the activity and thermal stability of cytolase PCL5, but also significantly improved the production of deapiose-xylosylated platycodin D from platycoside E. This bioconversion of platycosides enhanced by HHP could be applied to the industrial production of functional saponins.

**Author Contributions:** Conceptualization, C.S.N. and Y.-S.K.; methodology, K.-C.S., M.-J.S. and Y.J.O.; validation, D.W.K.; investigation, K.-C.S., Y.J.O. and D.W.K.; writing—original draft preparation, K.-C.S. and M.-J.S.; writing—review and editing, C.S.N. and Y.-S.K.; visualization, M.-J.S.; supervision, C.S.N. and Y.-S.K.; project administration, C.S.N. and Y.-S.K.; funding acquisition, C.S.N. All authors have read and agreed to the published version of the manuscript.

**Funding:** This study was carried out with the support of the R&D Program for Forest Science Technology (Project No. 2021400A00-2125-CA02) provided by the Korea Forest Service (Korea Forestry Promotion Institute).

**Institutional Review Board Statement:** Not applicable.

**Informed Consent Statement:** Not applicable.

**Acknowledgments:** This study was supported by the KU Research Professor Program of Konkuk University.

**Conflicts of Interest:** The authors declare no conflict of interest.

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
