# Peer review of "Improved Biotransformation of Platycoside E into Deapiose-Xylosylated Platycodin D by Cytolase PCL5 under High Hydrostatic Pressure"

_applsci, doi:10.3390/app112210623_

Round 1

Reviewer 1 Report

Authors evaluated the activity of Cytolase PLC5 in both AP and HHP conditions. Authors found the enzyme converted PE into Deapi-xyl-PD within 4hrs with higher productivity than that under AP. This is an interesting discovery, but I have some questions to be clarified.

Page 1, line 21: Authors wrote “increased the activity and stability of the enzyme by 5- and 4.9-fold”, and also in conclusion as “approximately 5-and 4.9-fold higher than those under AP”. But there is no documentation about “5-fold” data in results section. Please write “5-fold change” also in results section and show the data same manner.

Page 2, line 68: In “Enzyme assay”, I assume that this part described how to calculate substrate specificity. Please cite references or add explanation to describe author’s method in detail. I could not understand what author’s measure in Table 1. Did authors measure the concentration of substrate or products from substrate?

Page 3, line 90: In “HPLC Analysis”, authors used this system to evaluate the platycosides. I assume authors used standards of each platycosides. Please add explanation or cite references.

In Figure 5(b), I have an impression that PD3 showed no changes in this graph. If there was rapid increase and decrease of PD3 in this graph, please show the data another way to show the data clearly.

Author Response

Q1. Page 1, line 21: Authors wrote “increased the activity and stability of the enzyme by 5- and 4.9-fold”, and also in conclusion as “approximately 5-and 4.9-fold higher than those under AP”. But there is no documentation about “5-fold” data in results section. Please write “5-fold change” also in results section and show the data same manner.

Answer) Thank you for your concern. We have omitted to mention relevant content in the original manuscript. Therefore, a new sentence has been added as follows: “At 55 °C, the enzyme activity under HHP was 5-fold higher than that under AP.” (Line 123−124 of the revised manuscript)

Q2. Page 2, line 68: In “Enzyme assay”, I assume that this part described how to calculate substrate specificity. Please cite references or add explanation to describe author’s method in detail. I could not understand what author’s measure in Table 1. Did authors measure the concentration of substrate or products from substrate?

Answer) Thank you for your suggestion. As you suggested, we added an explanation of how to calculate specific activity as follows: “The specific activities of Cytolase PCL5 for platycosides such as platycoside E, platycodin D3, platycodin D, deapiosylated platycodin D, and deapiose-xylosylated platycodin D were evaluated at various concentrations (0.005–0.5 mg/mL) of the enzyme in order not to hydrolyze more than one sugar. The specific activity was defined as the amount of platycodin D3, platycodin D, deapiosylated platycodin D, or deapiose-xylosylated platycodin D, which was produced from platycoside E, platycodin D3, platycodin D, or deapiosylated platycodin D, respectively, as a product per enzyme amount per unit reaction time.” (Line 72−78 of the revised manuscript)

Q3. Page 3, line 90: In “HPLC Analysis”, authors used this system to evaluate the platycosides. I assume authors used standards of each platycosides. Please add explanation or cite references.

Answer) Thank you for your concern. We have already mentioned the related contents in the ‘Material’ part of the original manuscript as follows: “Platycoside E, platycodin D3, platycodin D, and deapiosylated platycodin D were purchased from Ambo Laboratories (Daejeon, Republic of Korea). Deapiose-xylosylated platycodin D was prepared as previously reported [22] and used as a standard.” (Line 64−66 of the original manuscript)

Instead, we added the calibration curves of standards in the ‘HPLC Analysis’ part of the revised manuscript as follows: “The calibration curves relating the logarithmic value of the peak areas to the concentrations of platycosides were drawnd using the solutions of platycoside standards (0.2 to 1.0 mM) and the curves were used to determine the concentrations of platycosides.” (Line 103−105 of the revised manuscript)

Q4. In Figure 5(b), I have an impression that PD3 showed no changes in this graph. If there was rapid increase and decrease of PD3 in this graph, please show the data another way to show the data clearly.

Answer) Thank you for your comment. Indeed, under HHP conditions, high hydrolytic activity made PD3 undetectable. Therefore, we added the following sentence to clarify this: “Under HHP condition, platycodin D3 was not detected throughout the reaction due to high hydrolytic activity.” (Line 165−166 of the revised manuscript)

Reviewer 2 Report

The manuscript entitled "Improved Biotransformation of Platycoside E into Deapiose-xylosylated Platycodin D by Cytolase PCL5 Under High Hydrostatic Pressure" represents new and interesting results concerning the biotransformation of platycoside. The purposes are clearly formulated and conclusions are justified by research results. The paper is well written and organized. In my opinion it can be published in the Applied Sciences journal.

Author Response

We would like to thank you for thoughtful review of this manuscript.

Reviewer 3 Report

Dear Authors,

The comments about your submitted work are pointed below:

The submitted paper “Improved Biotransformation of Platycoside E into Deapiose-xylosylated Platycodin D by Cytolase PCL5 under High Hydrostatic Pressure” of Kyung-Chun et al., reports on the improvement of the bioconversion of platycoside E, the most abundant saponin in balloon flower, by the use of the enzyme Cytolase PCL5 under high hydrostatic pressure.

In my opinion, the manuscript interestingly described the use of High Hydrostatic pressure to improve the enzymatic conversion of platycosides.

The manuscript is well written and the experimental section is described clearly and reproducibly.

The legends of the figures are clear and the figures are easily interpretable.

I only noticed that the paper should be implemented in the Introduction and in the Results and Discussion sections.

In the Introduction, I suggest to consider adding the progress/discovery on saponins, in particular platicosydes, and their importance in nutraceutical and pharmacological fields.

In my opinion, the Results and Discussion could be improved by implementing the section with comparisons of previous literature, which of course deals with the same subject.

Some points need to be clarified and justified are listed below:

Line 69-74: What is the assay performed to evaluate the enzymatic activity? Please specify and describe it, also indicating how the enzymatic activity was calculated.

Line 87-89: Please specify how many bioconversion tests have been carried out. Furthermore, it would be more complete to indicate how many samples were taken to be analyzed and in which time intervals.

Line 150: I believe there is an error in reporting the productivities of the enzyme conversion under AP. It is perhaps 66.7 µM/h. If so, please correct.

Line154-156: Are there other references and/or previous literature that could support the explanation given for the enzyme behavior? If so, please mention it.

Line 157-160: Please make sure that the legend is under the figure on the same page in the layout, for an easier and faster understanding.

Author Response

Q1. In the Introduction, I suggest to consider adding the progress/discovery on saponins, in particular platicosydes, and their importance in nutraceutical and pharmacological fields.

Answer) Thank you for your suggestion. 'They' in line 30 of the original manuscript meant platycosides, not balloon flowers. For better understanding, we revised the first paragraph of the introduction as follows: “Balloon flowers have been traditionally used as a medicinal diet in Northeast Asia, due to their efficacy against cold, cough, tonsillitis, sore throat, bronchitis, and chest congestion [1]. They contain saponins called platycosides, which have been attracting increasing interest among researchers because of their nutraceutical and pharmacological activities, including anti-inflammatory [2,3], anti-oxidant [4,5], antitumor [6,7], anti-obesity [8,9], and immunoregulatory effects [10,11].” (Line 29−33 of the revised manuscript)

Q2. In my opinion, the Results and Discussion could be improved by implementing the section with comparisons of previous literature, which of course deals with the same subject.

Answer) Thank you for your suggestion. As you suggested, we improved the 'Discussion' by comparing it to the previous literature as follows: “Even when HHP was applied to isoquercetin production with a-L-rhamnosidase in the previous study, it showed the highest productivity at 150 MPa [25]. However, it showed approximately 2.6-fold higher productivity than that under AP, which was lower than the increase in the hydrolytic activity of Cytolase PCL5.” (Line 112−115 of the revised manuscript) “which was higher than the 4-fold increase in thermal stability under HHP in study for the production of isoquercetin [25].” (Line 135−136 of the revised manuscript)

Q3. Line 69-74: What is the assay performed to evaluate the enzymatic activity? Please specify and describe it, also indicating how the enzymatic activity was calculated.

Answer) Thank you for your suggestion. As you suggested, we added an explanation of how to calculate specific activity as follows: “The specific activities of Cytolase PCL5 for platycosides such as platycoside E, platycodin D3, platycodin D, deapiosylated platycodin D, and deapiose-xylosylated platycodin D were evaluated at various concentrations (0.005–0.5 mg/mL) of the enzyme in order not to hydrolyze more than one sugar. The specific activity was defined as the amount of platycodin D3, platycodin D, deapiosylated platycodin D, or deapiose-xylosylated platycodin D, which was produced from platycoside E, platycodin D3, platycodin D, or deapiosylated platycodin D, respectively, as a product per enzyme amount per unit reaction time.” (Line 72−78 of the revised manuscript)

Q4. Line 87-89: Please specify how many bioconversion tests have been carried out. Furthermore, it would be more complete to indicate how many samples were taken to be analyzed and in which time intervals.

Answer) Thank you for your comment. We added a related sentence as follows: “Samples were taken at 5 min, 10 min, 30 min, 1 h, 3 h, 6 h, 9 h, 12 h, and 15 h under AP and at 5 min, 10 min, 30 min, 1 h, 2 h, 3 h, 4 h, and 5 h under HHP, respectively, and the experiment was performed in triplicate.” (Line 92−94 of the revised manuscript)

Q5. Line 150: I believe there is an error in reporting the productivities of the enzyme conversion under AP. It is perhaps 66.7 µM/h. If so, please correct.

Answer) Thank you for your pointing out. We miswrote the unit wrong in the original manuscript. Therefore, ‘%’ was deleted in the revised manuscript. (Line 162 of the revised manuscript)

Q6. Line154-156: Are there other references and/or previous literature that could support the explanation given for the enzyme behavior? If so, please mention it.

Answer) Thank you for your concern. We did not find any other reference that described the enzyme behavior presented in this study for stepwise hydrolysis. We explained this based on the substrate specificity in the original manuscript. Therefore, to clarify this, the related sentence was modified as follows: “During the time-course for reaction, the highest concentration of deapiosylated platycodin D was lower under HHP (467 mM at 1 h) than AP (609 mM at 6 h). Based on substrate specificity, this result may be attributable to the higher fold increase in the activity of Cytolase PCL5 towards deapiose-xylosylated platycodin D than that toward other platycoside intermediates.” (Line 165−169 of the revised manuscript)

Q7. Line 157-160: Please make sure that the legend is under the figure on the same page in the layout, for an easier and faster understanding.

Answer) Thank you for your concern. In the revised manuscript, the legend of Figure 5 was displayed on the same page along with the figure. (Line 170−173 of the revised manuscript)